# Exploring Prediction Uncertainty in Machine Translation Quality Estimation

## Abstract

Machine Translation Quality Estimation is a notoriously difficult task, which lessens its usefulness in real-world translation environments. Such scenarios can be improved if quality predictions are accompanied by a measure of uncertainty. However, models in this task are traditionally evaluated only in terms of point estimate metrics, which do not take prediction uncertainty into account. We investigate probabilistic methods for Quality Estimation that can provide well-calibrated uncertainty estimates and evaluate them in terms of their full posterior predictive distributions. We also show how this posterior information can be useful in an asymmetric risk scenario, which aims to capture typical situations in translation workflows.

## 1 Introduction

Quality Estimation (QE) (Blatz et al., 2004; Specia et al., 2009) models aim at predicting the quality of automatically translated text segments. Traditionally, these models provide point estimates and are evaluated using metrics like Mean Absolute Error (MAE), Root-Mean-Square Error (RMSE) and Pearson's $r$ correlation coefficient. However, in practice QE models are built for use in decision making in large workflows involving Machine Translation (MT). In these settings, relying on point estimates would mean that only very accurate prediction models can be useful in practice.

A way to improve decision making based on quality predictions is to explore uncertainty estimates. Consider for example a post-editing scenario where professional translators use MT in an effort to speed-up the translation process. A QE model can be used to determine if an MT segment is good enough for post-editing or should be discarded and translated from scratch. But since QE models are not perfect they can end up allowing bad MT segments to go through for post-editing because of a prediction error. In such a scenario, having an uncertainty estimate for the prediction can provide additional information for the filtering decision. For instance, in order to ensure good user experience for the human translator and maximise translation productivity, an MT segment could be forwarded for post-editing only if a QE model assigns a high quality score with *low uncertainty* (high confidence). Such a decision process is not possible with point estimates only.

Good uncertainty estimates can be acquired from well-calibrated probability distributions over the quality predictions. In QE, arguably the most successful probabilistic models are Gaussian Processes (GPs) since they considered the state-of-the-art for regression (Cohn and Specia, 2013; Hensman et al., 2013), especially in the low-data regimes typical for this task. We focus our analysis in this paper on GPs since other common models used in QE can only provide point estimates as predictions. Another reason why we focus on probabilistic models is because this lets us employ the ideas proposed by Quiñonero-Candela et al. (2006), which defined new evaluation metrics that take into account probability distributions over predictions.

The remaining of this paper is organized as follows:

- In Section 2 we further motivate the use of GPs for uncertainty modelling in QE and revisit their underlying theory. We also propose some model extensions previously developed in the GP literature and argue they are more appropriate for the task.

- We intrinsically evaluate our proposed models in terms of their posterior distributions on training and test data in Section 3. Specifically, we show that differences in uncertainty modelling are not captured by the usual point estimate metrics commonly used for this task.

- As an example of an application for predicitive distributions, in Section 4 we show how they can be useful in scenarios with asymmetric risk and how the proposed models can provide better performance in this case.

We discuss related work in Section 5 and give conclusions and avenues for future work in Section 6.

While we focus on QE as application, the methods we explore in this paper can be applied to any text regression task where modelling predictive uncertainty is useful, either in human decision making or by propagating this information for further computational processing.

## 2 Probabilistic Models for QE

Traditionally, QE is treated as a regression task with hand-crafted features. Kernel methods are arguably the state-of-the-art in QE since they can easily model non-linearities in the data. Furthermore, the scalability issues that arise in kernel methods do not tend to affect QE in practice since the datasets are usually small, in the order of thousands of instances.

The most popular method for QE is Support Vector Regression (SVR), as shown in the multiple instances of the WMT QE shared tasks (Callison-burch et al., 2012; Bojar et al., 2013; Bojar et al., 2014; Bojar et al., 2015). While SVR models can generate competitive predictions for this task, they lack a probabilistic interpretation, which makes it hard to extract uncertainty estimates using them. Bootstrapping approaches like bagging (Abe and Mamitsuka, 1998) can be applied, but this requires setting and optimising hyperparameters like bag size and number of bootstraps. There is also no guarantee these estimates come from a well-calibrated probabilistic distribution.

Gaussian Processes (GPs) (Rasmussen and Williams, 2006) is an alternative kernel-based framework that gives competitive results for point estimates (Cohn and Specia, 2013; Shah et al., 2013; Beck et al., 2014b). Unlike SVR, they explicitly model uncertainty in the data and in the predictions. This makes GPs very applicable when well-calibrated uncertainty estimates are required. Furthermore, they are very flexible in terms of modelling decisions by allowing the use of a variety of kernels and likelihoods while providing efficient ways of doing model selection. Therefore, in this work we focus on GPs for probabilistic modelling of QE. In what follows we briefly describe the GPs framework for regression.

### 2.1 Gaussian Process Regression

Here we follow closely the definition of GPs given by Rasmussen and Williams (2006). Let $\mathcal{X} = \{(\mathbf{x}_1, y_1), (\mathbf{x}_2, y_2), \ldots, (\mathbf{x}_n, y_n)\}$ be our data, where each $\mathbf{x} \in \mathbb{R}^D$ is a $D$-dimensional input and $y$ is its corresponding response variable. A GP is defined as a stochastic model over the latent function $f$ that generates the data $\mathcal{X}$:

$$f(\mathbf{x}) \sim \mathcal{GP}(m(\mathbf{x}), k(\mathbf{x}, \mathbf{x}')),$$

where $m(\mathbf{x})$ is the *mean* function, which is usually the $0$ constant, and $k(\mathbf{x}, \mathbf{x}')$ is the kernel or *covariance* function, which describes the covariance between values of $f$ at the different locations of $\mathbf{x}$ and $\mathbf{x}'$.

The prior is combined with a likelihood via Bayes' rule to obtain a posterior over the latent function:

$$p(f|\mathcal{X}) = \frac{p(\mathbf{y}|\mathbf{X}, f)p(f)}{p(\mathbf{y}|\mathbf{X})},$$

where $\mathbf{X}$ and $\mathbf{y}$ are the training inputs and response variables, respectively. For regression, we assume that each $y_i = f(\mathbf{x_i}) + \eta$, where $\eta \sim \mathcal{N}(0, \sigma_n^2)$ is added white noise. Having a Gaussian likelihood results in a closed form solution for the posterior.

Training a GP involves the optimisation of model hyperparameters, which is done by maximising the marginal likelihood $p(\mathbf{y}|\mathbf{X})$ via gradient ascent. Predictive posteriors for unseen $\mathbf{x}_*$ are obtained by integrating over the latent function evaluations at $\mathbf{x}_*$.

GPs can be extended in many different ways by applying different kernels, likelihoods and modifying the posterior, for instance. In the next Sections, we explain in detail some sensible modelling choices in applying GPs for QE.

### 2.2 Matèrn Kernels

Choosing an appropriate kernel is a crucial step in defining a GP model (and any other kernel

method). A common choice is to employ the exponentiated quadratic (EQ) kernel[1]:

$$k_{\text{EQ}}(\mathbf{x}, \mathbf{x}') = \sigma_v \exp\left(-\frac{r^2}{2}\right),$$

$$\text{where } r^2 = \sum_{i=1}^{D} \frac{(x_i - x_i')^2}{l_i^2}$$

is the scaled distance between the two inputs, $\sigma_v$ is a scale hyperparameter and $\mathbf{l}$ is a vector of lengthscales. Most kernel methods tie all lengthscale to a single value, resulting in an isotropic kernel. However, since in GPs hyperparameter optimisation can be done efficiently, it is common to employ one lengthscale per feature, a method called Automatic Relevance Determination (ARD).

The EQ kernel allows the modelling of non-linearities between the inputs and the response variables but it makes a strong assumption: it generates smooth, infinitely differentiable functions. This assumption can be too strong for noisy data. An alternative is the Matèrn class of kernels, which relax the smoothness assumption by modelling functions which are $\nu$-times differentiable only. Common values for $\nu$ are the half-integers $3/2$ and $5/2$, resulting in the following Matèrn kernels:

$$k_{\text{M32}} = \sigma_v(1 + \sqrt{3r^2}) \exp(-\sqrt{3r^2})$$

$$k_{\text{M52}} = \sigma_v \left(1 + \sqrt{5r^2} + \frac{5r^2}{3}\right) \exp(-\sqrt{5r^2}),$$

where we have omitted the dependence of $k_{\text{M32}}$ and $k_{\text{M52}}$ on the inputs $(\mathbf{x}, \mathbf{x}')$ for brevity. Higher values for $\nu$ are usually not very useful since the resulting behaviour is hard to distinguish from limit case $\nu \to \infty$, which retrieves the EQ kernel (Rasmussen and Williams, 2006, Sec. 4.2).

The relaxed smoothness assumptions from the Matèrn kernels makes them promising candidates for QE datasets, which tend to be very noisy. We expect that employing them will result in a better models for this application.

## 2.3 Warped Gaussian Processes

The Gaussian likelihood of standard GPs has support over the entire real number line. However, common quality scores are strictly positive values, which means that the Gaussian assumption

is not ideal. A usual way to deal with this problem is model the logarithm of the response variables, since this transformation maps strictly positive values to the real line. However, there is no reason to believe this is the best possible mapping: a better idea would be to learn it from the data.

Warped GPs (Snelson et al., 2004) are an extension of GPs that allows the learning of arbitrary mappings. It does that by placing a monotonic *warping function* over the observations and modelling the warped values inside a standard GP. The posterior distribution is obtained by applying a change of variables:

$$p(y_*|\mathbf{x}_*) = \frac{f'(y_*)}{\sqrt{2\pi\sigma_*^2}} \exp\left(\frac{f(y_*) - \mu_*}{2\sigma_*}\right),$$

where $\mu_*$ and $\sigma_*$ are the mean and standard deviation of the latent (warped) response variable and $f$ and $f'$ are the warping function and its derivative.

Point predictions from this model depend on the loss function to be minimised. For absolute error, the median is the optimal value while for squared error it is the mean of the posterior. In standard GPs, since the posterior is Gaussian the median and mean coincide but this in general is not the case for a Warped GP posterior. The median can be easily obtained by applying the inverse warping function to the latent median:

$$y_*^{\text{med}} = f^{-1}(\mu_*).$$

While the inverse of the warping function is usually not available in closed form, we can use its gradient to have a numerical estimate.

The mean is obtained by integrating $y^*$ over the latent density:

$$\mathbb{E}[y_*] = \int f^{-1}(z)\mathcal{N}_z(\mu_*, \sigma_*^2)dz,$$

where $z$ is the latent variable. This can be easily approximated using Gauss-Hermite quadrature since it is a one dimensional integral over a Gaussian density.

The warping function should be flexible enough to allow the learning of complex mappings, but it needs to be monotonic. Snelson et al. (2004) proposes a parametric form composed of a sum of $\tanh$ functions, similar to a neural network layer:

$$f(y) = y + \sum_{i=1}^{I} a_i \tanh(b_i(y + c_i)),$$

---

[1] Also known as Radial Basis Function (RBF) kernel.

where $I$ is the number of $\tanh$ terms and $\mathbf{a}$, $\mathbf{b}$ and $\mathbf{c}$ are treated as model hyperparameters and optimised jointly with the kernel and likelihood hyperparameters. Large values for $I$ allow more complex mappings to be learned but raise the risk of overfitting.

Warped GPs provide an easy and elegant way to model response variables with non-Gaussian behaviour within the GP framework. In our experiments we explore models employing warping functions with up to 3 terms, which is the value recommended by Snelson et al. (2004). We also report results using the $f(y) = \log(y)$ warping function.

## 3 Intrinsic Uncertainty Evaluation

Given a set of different probabilistic QE models, we are interested in evaluating the performance of these models, while also taking their uncertainty into account, particularly to distinguish among models with seemingly same or similar performance. A straightforward way to measure the performance of a probabilistic model is to inspect its negative ($\log$) marginal likelihood. This measure, however, does not capture if a model overfit the training data.

We can have a better generalization measure by calculating the likelihood on *test data* instead. This was proposed in previous work and it is called Negative Log Predictive Density (NLPD) (Quiñonero-Candela et al., 2006):

$$\text{NLPD}(\hat{\mathbf{y}}, \mathbf{y}) = -\frac{1}{n} \sum_{i=1}^{n} \log p(\hat{y}_i = y_i | \mathbf{x}_i).$$

where $\hat{\mathbf{y}}$ is a set of test predictions, $\mathbf{y}$ is the set of true labels and $n$ is the test set size. This metric has since been largely adopted by the ML community when evaluating GPs and other probabilistic models for regression (see Section 5 for some examples).

As with other error metrics, lower values are better. Intuitively, if two models produce equally incorrect predictions but they have different uncertainty estimates, NLPD will penalise the overconfident model more than the underconfident one. On the other hand, if predictions are close to the true value then NLPD will penalise the underconfident model instead.

In our first set of experiments we evaluate models proposed in Section 2 according to their negative log likelihood (NLL) and the NLPD on test data. We also report two point estimate metrics on test data: Mean Absolute Error (MAE), the most commonly used evaluation metric in QE, and Pearson's $r$, which has recently proposed by Graham (2015) as a more robust alternative.

### 3.1 Experimental Settings

Our experiments comprise datasets containing three different language pairs, where the label to predict is post-editing time:

**English-Spanish (en-es)** This dataset was used in the WMT14 QE shared task (Bojar et al., 2014). It contains $858$ sentences translated by one MT system and post-edited by a professional translator.

**French-English (fr-en)** Described in (Specia, 2011), this dataset contains $2,525$ sentences translated by one MT system and post-edited by a professional translator.

**English-German (en-de)** This dataset is part of the WMT16 QE shared task[2]. It was translated by one MT system for consistency we use a subset of $2,828$ instances post-edited by a single professional translator.

As part of the process of creating these datasets, post-editing time was logged on an sentence basis for all datasets. Following common practice, we normalise the post-editing time by the length of the machine translated sentence to obtain post-editing *rates* and use these as our response variables. For model building, we use a standard set of 17 features from the QuEst framework (Specia et al., 2015). These features are used in the strong baseline models provided by the WMT QE shared tasks. While the best performing systems in the shared tasks use larger feature sets, these are mostly resource-intensive and language-dependent, and therefore not equally applicable to all our language pairs. Moreover, our goal is to compare probabilistic QE models through the predictive uncertainty perspective, rather than improving the state-of-the-art in terms of point predictions. We perform 10-fold cross validation instead of using a single train/test splits and report averaged metric scores.

The model hyperparameters were optimised by maximising the likelihood on the training data. We perform a two-pass procedure similar to that

---

[2]www.statmt.org/wmt16

in (Cohn and Specia, 2013): first we employ an isotropic kernel and optimise all hyperparameters using 10 random restarts; then we move to an ARD equivalent kernel and perform a final optimisation step to fine tune feature *lengthscales*. Point predictions were fixed as the median of the distribution.

## 3.2 Results and Discussion

Table 1 shows the results obtained for all datasets. The first two columns shows an interesting finding in terms of model learning: using a warping function drastically decreases both NLL and NLPD. The main reason behind this is that standard GPs distribute probability mass over negative values, while the warped models do not. For the **fr-en** and **en-de** datasets, NLL and NLPD follow similar trends. This means that we can trust NLL as a measure of uncertainty for these datasets. However, this is not observed in the **en-es** dataset. Since this dataset is considerably smaller than the others, we believe this is evidence of overfitting, thus showing that NLL is not a reliable metric for small datasets.

In terms of different warping functions, using the parametric $\tanh$ function with 3 terms performs better than the $\log$ for the **fr-en** and **en-de** datasets. This is not the case of the **en-es** dataset, where the $\log$ function tends to perform better. We believe that this is again due to the smaller dataset size. The gains from using a Matèrn kernel over EQ are less conclusive. While they do tend to perform better for **fr-en**, there does not seem to be any difference in the other datasets. Different kernels might be more appropriate depending on the language pair but more experiments are needed to check if this is the case, which we leave for future work.

The differences in uncertainty modelling are by and large not captured by the point estimate metrics. While MAE does show gains from standard to Warped GPs, it does not reflect the difference found between warping functions for **fr-en**. Pearson's $r$ is also quite inconclusive in this sense, except for some observed gains for **en-es**. This shows that NLPD indeed should be preferred as a evaluation metric when proper prediction uncertainty estimates are required by a QE model.

### English-Spanish - 858 instances

|  | NLL | NLPD | MAE | $r$ |
|---|---|---|---|---|
| EQ | 1244.03 | 1.632 | 0.828 | 0.362 |
| Mat32 | 1237.48 | 1.649 | 0.862 | 0.330 |
| Mat52 | 1240.76 | 1.637 | 0.853 | 0.340 |
| log EQ | 986.14 | 1.277 | 0.798 | 0.368 |
| log Mat32 | 982.71 | 1.271 | 0.793 | 0.380 |
| log Mat52 | 982.31 | 1.272 | 0.794 | 0.376 |
| tanh1 EQ | 992.19 | 1.274 | 0.790 | 0.375 |
| tanh1 Mat32 | 991.39 | 1.272 | 0.790 | 0.379 |
| tanh1 Mat52 | 992.20 | 1.274 | 0.791 | 0.376 |
| tanh2 EQ | 982.43 | 1.275 | 0.792 | 0.376 |
| tanh2 Mat32 | 982.40 | 1.281 | 0.791 | 0.382 |
| tanh2 Mat52 | 981.86 | 1.282 | 0.792 | 0.278 |
| tanh3 EQ | 980.50 | 1.282 | 0.791 | 0.380 |
| tanh3 Mat32 | 981.20 | 1.282 | 0.791 | 0.380 |
| tanh3 Mat52 | 980.70 | 1.275 | 0.790 | 0.385 |

### French-English - 2525 instances

|  | NLL | NLPD | MAE | $r$ |
|---|---|---|---|---|
| EQ | 2334.17 | 1.039 | 0.491 | 0.322 |
| Mat32 | 2335.81 | 1.040 | 0.491 | 0.320 |
| Mat52 | 2344.86 | 1.037 | 0.490 | 0.320 |
| log EQ | 1935.71 | 0.855 | 0.493 | 0.314 |
| log Mat32 | 1949.02 | 0.857 | 0.493 | 0.310 |
| log Mat52 | 1937.31 | 0.855 | 0.493 | 0.313 |
| tanh1 EQ | 1884.82 | 0.840 | 0.482 | 0.322 |
| tanh1 Mat32 | 1890.34 | 0.840 | 0.482 | 0.317 |
| tanh1 Mat52 | 1887.41 | 0.834 | 0.482 | 0.320 |
| tanh2 EQ | 1762.33 | 0.775 | 0.483 | 0.323 |
| tanh2 Mat32 | 1717.62 | 0.754 | 0.483 | 0.313 |
| tanh2 Mat52 | 1748.62 | 0.768 | 0.486 | 0.306 |
| tanh3 EQ | 1814.99 | 0.803 | 0.484 | 0.314 |
| tanh3 Mat32 | 1723.89 | 0.760 | 0.486 | 0.302 |
| tanh3 Mat52 | 1706.28 | 0.751 | 0.482 | 0.320 |

### English-German - 2828 instances

|  | NLL | NLPD | MAE | $r$ |
|---|---|---|---|---|
| EQ | 4852.80 | 1.865 | 1.103 | 0.359 |
| Mat32 | 4850.27 | 1.861 | 1.098 | 0.369 |
| Mat52 | 4850.33 | 1.861 | 1.098 | 0.369 |
| log EQ | 4053.43 | 1.581 | 1.063 | 0.360 |
| log Mat32 | 4054.51 | 1.580 | 1.063 | 0.363 |
| log Mat52 | 4054.39 | 1.581 | 1.064 | 0.363 |
| tanh1 EQ | 4116.86 | 1.597 | 1.068 | 0.343 |
| tanh1 Mat32 | 4113.74 | 1.593 | 1.064 | 0.351 |
| tanh1 Mat52 | 4112.91 | 1.595 | 1.068 | 0.349 |
| tanh2 EQ | 4032.70 | 1.570 | 1.060 | 0.359 |
| tanh2 Mat32 | 4031.42 | 1.570 | 1.060 | 0.362 |
| tanh2 Mat52 | 4032.06 | 1.570 | 1.060 | 0.361 |
| tanh3 EQ | 4023.72 | 1.569 | 1.062 | 0.359 |
| tanh3 Mat32 | 4024.64 | 1.567 | 1.058 | 0.364 |
| tanh3 Mat52 | 4026.07 | 1.566 | 1.059 | 0.365 |

Table 1: Intrinsic evaluation results. The first three rows in each table correspond to standard GP models, while the remaining rows are Warped GP models with different warping functions. The number after the "tanh" models shows the number of terms in the warping function (see Equation 2.3). All $r$ scores have $p < 0.05$.

## 4 Asymmetric Risk Scenarios

Evaluation metrics for QE, including those used in the WMT QE shared tasks, are assumed to be symmetric, i.e., they penalise over and underestimates equally. This assumption is however too simplistic for many possible applications of QE. For example:

- In a *post-editing* scenario, a project manager may have translators with limited expertise in post-editing. In this case, automatic translations should not be provided to the translator unless they are highly likely to have very good quality. This can be enforced this by increasing the penalisation weight for underestimates. We call this the *pessimistic* scenario.

- In a *gisting* scenario, a company wants to automatically translate their product reviews so that they can be published in a foreign language without human intervention. The company would prefer to publish only the reviews translated well enough, but having more reviews published will increase the chances of selling products. In this case, having better recall is more important and thus only reviews with very poor translation quality should be discarded. We can accomplish this by heavier penalisation on overestimates, a scenario we call *optimistic*.

In this Section we show how these scenarios can be addressed by well-calibrated predictive distributions and by employing *asymmetric* loss functions. An example of such a function is the asymmetric linear (henceforth, AL) loss, which is a generalisation of the absolute error:

$$L(\hat{y}, y) = \begin{cases} w(\hat{y} - y) & \text{if } \hat{y} > y \\ y - \hat{y} & \text{if } \hat{y} \leq y, \end{cases}$$

where $w > 0$ is the weight given to overestimates. If $w > 1$ we have the pessimistic scenario, and the optimistic one can be obtained using $0 < w < 1$. For $w = 1$ we retrieve the original absolute error loss.

Another asymmetric loss is the linear exponential or *linex* loss (Zellner, 1986):

$$L(\hat{y}, y) = \exp[w(\hat{y} - y)] - (\hat{y} - y) - 1$$

where $w \in \mathbb{R}$ is the weight. This loss attempts to keep a linear penalty in lesser risk regions, while

imposing an exponential penalty in the higher risk ones. Negative values for $w$ will result in a pessimistic setting, while positive values will result in the optimistic one. For $w = 0$, the loss approximates a squared error loss. Usual values for $w$ tend to be close to $1$ or $-1$ since for higher weights the loss can quickly reach very large scores. Both losses are shown on Figure 1.

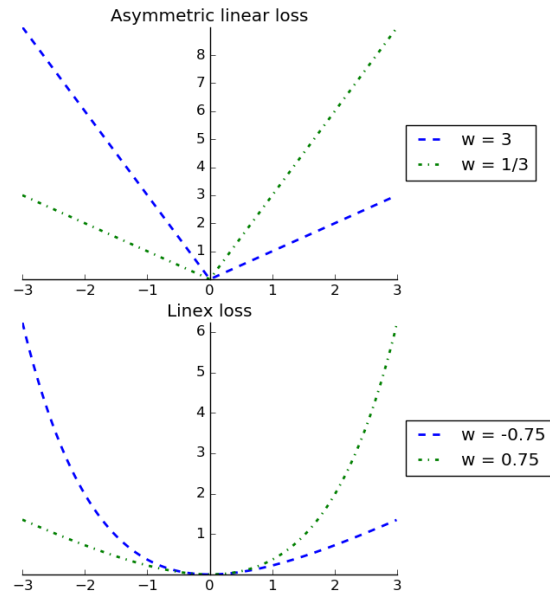

Figure 1: Asymmetric losses. The blue lines correpond to the pessimistic scenario since it imposes larger penalties when the prediction is lower than the true label. Conversely, the red lines represent the optimistic scenario.

### 4.1 Bayes Risk for Asymmetric Losses

The losses introduced above can be incorporated directly into learning algorithms to obtain models for a given scenario. In the context of the AL loss this is called *quantile regression* (Koenker, 2005), since optimal estimators for this loss are posterior quantiles. However, in a production environment the loss can change over time. For instance, in the gisting scenario discussed above the parameter $w$ could be changed based on feedback from indicators of sales revenue or user experience. If the loss is attached to the underlying learning algorithms, a change in $w$ would require full model retraining, which can be costly.

Instead of retraining the model everytime there is a different loss, we can train a single probabilistic model and derive Bayes risk estimators for the loss we are interested in. This allows estimates to

be obtained without having to retrain models when the loss changes. Additionally, this allows different losses/scenarios to be employed at the same time using the same model.

Minimum Bayes risk estimators for asymmetric losses were proposed by Christoffersen and Diebold (1997) and we follow their derivations in our experiments. The best estimator for the AL loss is equivalent to the $\frac{w}{w+1}$ quantile of the predictive distribution. Note that we retrieve the median when $w = 1$, as expected. The best estimator for the linex loss can be easily derived and results in:

$$\hat{y} = \mu_y - \frac{w\sigma_y^2}{2}$$

where $\mu_y$ and $\sigma_y^2$ are the mean and the variance of the predictive posterior.

### 4.2 Experimental Settings

Here we assess the models and datasets used in Section 3.1 in terms of their performance in the asymmetric setting. Following the explanation in the previous Section, we do not perform any retraining: we collect the predictions obtained using the 10-fold cross-validation protocol and apply different Bayes estimators corresponding to the asymmetric losses. Evaluation is performed using the same loss employed in the estimator (for instance, when using the linex estimator with $w = 0.75$ we report the results using the linex loss with same $w$) and averaged over the 10 folds.

To simulate both pessimistic and optimistic scenarios, we use $w \in \{3, 1/3\}$ for the AL loss and $w \in \{-0.75, 0.75\}$ for the linex loss. The only exception is the **en-de** dataset, where we report results for $w \in -0.25, 0.75$ for linex[3]. We also report results only for models using the Matèrn52 kernel. While we did experiment with different kernels and weighting schemes[4] our findings showed similar trends so we omit them for the sake of clarity.

### 4.3 Results and Discussion

Results are shown on Table 2. In the optimistic scenario the tanh-based warped GP models give

---

[3]Using $w = -0.75$ in this case resulted in loss values on the order of $10^7$. In fact, as it will be discussed in the next Section, the results for the linex loss in the pessimistic scenario were inconclusive. However, we report results using a higher $w$ in this case for completeness and to clarify the inconclusive trends we found.

[4]We also tried $w \in \{1/9, 1/7, 1/5, 5, 7, 9\}$ for the AL loss and $w \in \{-0.5, -0.25, 0.25, 0.5\}$ for the linex loss.

**English-Spanish**

| | Optimistic | | Pessimistic | |
|---|---|---|---|---|
| | AL | Linex | AL | Linex |
| Std GP | 1.187 | 0.447 | 1.633 | 3.009 |
| log | 1.060 | 0.299 | 1.534 | 3.327 |
| tanh1 | 1.050 | 0.300 | 1.528 | 3.251 |
| tanh2 | 1.054 | 0.300 | 1.543 | 3.335 |
| tanh3 | 1.053 | 0.299 | 1.538 | 3.322 |

**French-English**

| | Optimistic | | Pessimistic | |
|---|---|---|---|---|
| | AL | Linex | AL | Linex |
| Std GP | 0.677 | 0.127 | 0.901 | 0.337 |
| log | 0.675 | 0.161 | 0.914 | 0.492 |
| tanh1 | 0.677 | 0.124 | 0.901 | 0.341 |
| tanh2 | 0.671 | 0.121 | 0.894 | 0.347 |
| tanh3 | 0.666 | 0.120 | 0.886 | 0.349 |

**English-German**

| | Optimistic | | Pessimistic | |
|---|---|---|---|---|
| | AL | Linex | AL | Linex |
| Std GP | 1.528 | 0.610 | 2.120 | 0.217 |
| log | 1.457 | 0.537 | 2.049 | 0.222 |
| tanh1 | 1.459 | 0.503 | 2.064 | 0.220 |
| tanh2 | 1.455 | 0.504 | 2.045 | 0.220 |
| tanh3 | 1.456 | 0.497 | 2.042 | 0.219 |

Table 2: Asymmetric loss experiments results. The first line in each table corresponds to a standard GP while the others are Warped GPs with different warping functions. All models use the Matèrn52 kernel. The optimistic setting corresponds to $w = 1/3$ for AL and $w = 0.75$ for linex. The pessimistic setting uses $w = 3$ for AL and $w = -0.75$ for linex, except for English-German, where $w = -0.25$.

consistently better results than standard GPs. The log-based models also gives good results for AL but for Linex the results are mixed except for en-es. This is probably again related to the larger sizes of the fr-en and en-de datasets, which allows the tanh-based models to learn richer representations.

The pessimistic scenario shows interesting trends. While the results for AL follow a similar pattern when compared to the optimistic setting, the results for linex are consistently worse than the standard GP baseline. A key difference between AL and linex is that the latter depends on the

variance of the predictive distribution. Since the warped models tend to have less variance, we believe the estimator is not being "pushed" towards the positive tails as much as in the standard GPs. This turns the resulting predictions not conservative enough (i.e. the post-editing time predicitions are lower) and this is heavily (exponentially) penalized by the loss. This might be a case where a standard GP is preferred but can also indicate that this loss is biased towards models with high variance, even if it does that by assigning probability mass to nonsensical values (like negative time). We leave further investigation of this phenomenon for future work.

## 5 Related Work

Quality Estimation is generally framed as text regression task, similarly to many other applications such as movie revenue forecasting based on reviews (Joshi et al., 2010; Bitvai and Cohn, 2015) and detection of emotion strength in news headlines (Strapparava and Mihalcea, 2008; Beck et al., 2014a) and song lyrics (Mihalcea and Strapparava, 2012). In general, these applications are evaluated in terms of their point estimate predictions, arguably because not all of them employ probabilistic models.

The NLPD is common and established metric used in the GP literature to evaluate new approaches. Examples include the original work on Warped GPs (Snelson et al., 2004), but also others like Lázaro-Gredilla (2012) and Chalupka et al. (2013). It has also been used to evaluate recent work on uncertainty propagation methods for neural networks (Hernández-Lobato and Adams, 2015).

Asymmetric loss functions are common in the econometrics literature and were studied by Zellner (1986) and Koenker (2005), among others. Besides the AL and the linex, another well studied loss is the asymmetric quadratic, which in turn relates to the concept of *expectiles* (Newey and Powell, 1987). This loss generalises the commonly used squared error loss. In terms of applications, Cain and Janssen (1995) gives an example in real estate assessment, where the consequences of under and overassessment are usually different depending on the specific scenario. An engineering example is given by Zellner (1986) in the context of dam construction, where an underestimate of peak water level is much more serious than an overestimate. Such real-world applications guided many developments in this field: we believe that translation and other language processing scenarios which rely on NLP technologies can heavily benefit from these advancements.

## 6 Conclusions

This work explored new probabilistic models for machine translation QE that allow better uncertainty estimates. We proposed the use of NLPD, which can capture information on the whole predictive distribution, unlike usual point estimate-based metrics. By assessing models using NLPD we can make better informed decisions about which model to employ for different settings. Furthermore, we showed how information in the predictive distribution can be used in asymmetric loss scenarios and how the proposed models can be beneficial in these settings.

Uncertainty estimates can be useful in many other settings beyond the ones explored in this work. Active Learning can benefit from variance information in their query methods and it has shown to be useful for QE (Beck et al., 2013). Exploratory analysis is another avenue for future work, where error bars can provide further insights about the task, as shown in recent work (Nguyen and O'Connor, 2015). This kind of analysis can be useful for tracking post-editor behaviour and assessing cost estimates for translation projects, for instance.

Our main goal in this paper was to raise awareness about how different modelling aspects should be taken into account when building QE models. Decision making can be risky using simple point estimates and we believe that uncertainty information can be beneficial in such scenarios by providing more informed solutions. These ideas are not restricted to QE and we hope to see similar studies in other natural language applications in the future.

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
