# Peer review of "Exploring Prediction Uncertainty in Machine Translation Quality Estimation"

_CoNLL 2016 — decision unknown_

[Official Review · Reviewer 1 · rating 4 · confidence 3]
soundness 5 · originality 3 · clarity 4 · impact 3 · substance 4 · appropriateness 5 · meaningful comparison 5 · replicability 4 · presentation format Poster

The paper explores the use of probabilistic models (gaussian processes) to
regress on the target variable of post-editing time/rates for quality
estimation of MT output.
The paper is well structured with a clear introduction that highlights the
problem of QE point estimates in real-world applications. I especially liked
the description of the different asymmetric risk scenarios and how they entail
different estimators.
For readers familiar with GPs the paper spends quite some space to reflect
them, but I think it is worth the effort to introduce these concepts to the
reader.
The GP approach and the choices for kernels and using warping are explained
very clearly and are easy to follow. In general the research questions that are
to be answered by this paper are interesting and well phrased.

However, I do have some questions/suggestions about the Results and Discussion
sections for Intrinsic Uncertainty Evaluation:
- Why were post-editing rates chosen over prediction (H)TER? TER is a common
value to predict in QE research and it would have been nice to justify the
choice made in the paper.
- Section 3.2: I don't understand the first paragraph at all: What exactly is
the trend you see for fr-en & en-de that you do not see for en-es? NLL and NLPD
'drastically' decrease with warped GPs for all three datasets.
- The paper indeed states that it does not want to advance state-of-the-art
(given that they use only the standard 17 baseline features), but it would have
been nice to show another point estimate model from existing work in the result
tables, to get a sense of the overall quality of the models.
- Related to this, it is hard to interpret NLL and NLPD values, so one is
always tempted to look at MAE in the tables to get a sense of 'how different
the predictions are'. Since the whole point of the paper is to say that this is
not the right thing to do, it would be great provide some notion of what is a
drastic reduce in NLL/NLPD worth: A qualitative analysis with actual examples.

Section 4 is very nicely written and explains results very intuitively!

Overall, I like the paper since it points out the problematic use of point
estimates in QE. A difficult task in general where additional information such
as confidence arguably are very important. The submission does not advance
state-of-the-art and does not provide a lot of novelty in terms of modeling
(since GPs have been used before), but its research questions and goals are
clearly stated and nicely executed.

Minor problems:
- Section 4: "over and underestimates" -> "over- and underestimates"
- Figure 1 caption: Lines are actually blue and green, not blue and red as
stated in the caption.
- If a certain toolkit was used for GP modeling, it would be great to refer to
this in the final paper.